# *Look Ma, No Training!* OBSERVATION SPACE DESIGN FOR REINFORCEMENT LEARNING

## ABSTRACT

Many scientific communities agree on the potential of reinforcement learning (RL) agents to solve real-world problems, yet such consensus does not extend to how these agents should be designed. In some practical applications, the increasing literature on RL does not shed light on which RL components work better for a particular problem, they are usually treated just as configuration elements to be reported. One of these components is the choice of observation space, which in some cases entails dealing with tens of thousands of observable features. Choosing a rich yet efficient observation space is key to encoding useful information while limiting the tangible implications of adding extra features. Gaining understanding of feature relevance has already been studied for RL. In comparison to supervised learning, the effect of dependencies across states adds a layer of complexity to the structure of the problem. Many of the proposed methods require training RL agents from scratch several times, which is costly in real-world applications. In this paper we propose a simple and cost-efficient way to find good observation spaces that does not require training. Specifically, we propose leveraging multiple random policies when comparing candidate spaces for the same problem. By conducting rollouts with different random policies for each candidate space, we are able to identify statistically-significant signals that indicate which features are better suited for the application considered. We demonstrate the usefulness of our approach in different RL problems, including Traffic Signal Control. By combining random policy sampling with the Hill Climbing search algorithm, we find observation spaces that use less features and achieve comparable or greater return. Overall, this work suggests a straightforward and inexpensive approach to an important aspect of RL design that is often overlooked and is crucial for applied problems.

## 1   INTRODUCTION

Reinforcement Learning (RL) is a rapidly growing field with the potential to solve a wide range of real-world problems. Even though progress is reported in the literature, it seems that the impact in practical applications is currently limited (Henderson et al., 2018; Andrychowicz et al., 2020; Parker-Holder et al., 2022) and mostly occurs when a large amount of computational resources and human expertise is available (Ceron & Castro, 2021). One aspect of this bottleneck is the little understanding of optimal RL design for real-world problems. In many cases, the literature does not provide clear guidance on which components work best for a specific problem and consensus does not naturally emerge either.

One of these key RL components is the observation space. The choice of observation space has a direct impact on the performance of the agent (Reda et al., 2020; Singh et al., 2020; Kim & Ha, 2021) and it often consists of hand-crafted sets of features. The literature around concrete applications demonstrates that in many cases there is no convergence on which feature sets to use. For example, recent studies on RL for Traffic Signal Control (TSC) (Wei et al., 2021; Higuera et al., 2019; Zhou et al., 2019) propose more than 10 different feature sets to solve the same problem (Noaeen et al., 2022). Another common approach to observation space design is using feature sets of high dimensions (e.g., adding features for completion, for symmetry, etc). While adding extra features may not confer any additional advantage (Guyon & Elisseeff, 2003), it may not be desirable from a practical perspective either, as every extra feature may entail additional data that must be collected, which may be more intrusive for users or incur substantial hardware costs.

Reducing the dimensionality of feature sets by identifying the most useful features is a long-studied problem in optimization and Machine Learning (ML) (Hall, 1999; Chandrashekar & Sahin, 2014; Cai et al., 2018). Its application in RL adds a layer of complexity, since the impact of a feature at a certain timestep might not be obvious until a subsequent timestep in the future. Therefore, any feature selection strategy must account for these temporal dependencies. Different approaches have been proposed to address this in RL (Bertsekas et al., 1988; Keller et al., 2006; Parr et al., 2008; Liu et al., 2012; Shen & Chi, 2016; Liu et al., 2021); in the majority of the cases they require training the RL agent many times. This approach is not sustainable in real-world RL problems, since training multiple times is costly and other components must be also designed during the search process.

In this paper we address this gap by proposing an observation space design method that is cost-efficient and straightforward to use. Specifically, we argue that sampling random policies and evaluating them in the environment through multiple rollouts is informative and can help in identifying which features are relevant. The outcome of this process is a distribution of returns; when comparing such distribution for different candidate observation spaces, statistically significant differences can be observed. The idea of random sampling is not new in ML (Bergstra & Bengio, 2012) nor in RL (Mania et al., 2018; Barrett et al., 2020), although its potential remains unexplored. We demonstrate that, combined with a search algorithm like Hill Climbing, it offers a powerful yet inexpensive method of finding optimal observation spaces or, at least, greatly reduce the set of candidates.

We ground our analyses in different RL problems, including TSC. We first derive theoretical intuition behind random policy sampling and present a toy example with random linear policies. Then, we propose an algorithm that integrates Hill Climbing search with random policy sampling and test it on the Paddle environment (Verma, 2020) and the RESCO benchmark in TSC (Eom & Kim, 2020). Without any kind of training, our approach finds observation spaces that use less features. Then, after using these observation spaces in training runs, these achieve better returns. While the strength of our approach is its simplicity and cost-efficiency, we observe that, in some cases, the stochasticity of this approach entails no guarantee of finding the optimal observation space. Still, our method always identifies a subset of promising candidates, which is very valuable in real-world RL. We hope this work serves as a stepping stone towards improving observation space design in RL and adapting feature selection mechanisms to the unique context of RL.

## 2 LIMITATIONS OF CURRENT OBSERVATION SPACE DESIGN STRATEGIES

We begin by motivating the study of observation space design in RL. In this section we discuss two limitations of many of current observation spaces: an excess of features, and literature not showing signs of practitioners converging to uniform representations.

### 2.1 EXCESS OF FEATURES

Designing an observation representation is frequently an empirical process; when RL agents do not work with raw sensor inputs, this process typically involves selecting features that are relevant to the problem at hand. When constructing these hand-crafted features, a common approach is to include as many features as possible, as it increases the amount of information available to the agent, hoping to achieve better performance.

However, we observe that providing too much information to the agent in the form of an excess of features can degrade performance. We show this effect in Figure 1, in which we run the same experiment across three different robotics environments from PyBullet (Coumans & Bai, 2016). In each experiment, we take the default state space given by the environments' codebases, which is composed by features encoding positional and inertial information of the robot body and its joints, and mask out groups of features one group at a time. We use PPO (Schulman et al., 2017) and a fixed configuration (see Appendix A) to train seven different agents from scratch, each using a different feature mask, and compare the resulting performance with an eighth agent that uses all features.

In all cases, we observe that using all features does not confer any additional advantage over masking out a group of features and sometimes it actually leads to a worse result. These outcomes align with similar experiments conducted by Reda et al. (2020) and Kim & Ha (2021). In addition, even though the environments share certain dynamics, the features that are beneficial and detrimental in each environment are different (e.g., masking out the XYZ body velocity helps in Ant but not in Hopper).

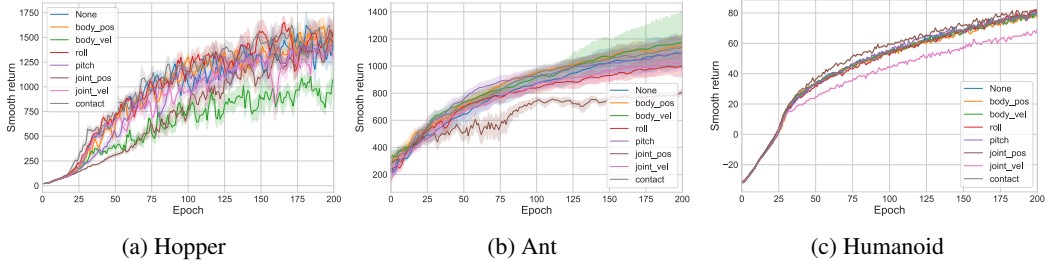

| (a) Hopper | (b) Ant | (c) Humanoid |

Figure 1: Training curves in three different PyBullet environments comparing a complete observation space (*None*) with subspaces in which we mask one of the following elements: XYZ position of the body, XYZ velocity of the body, roll, pitch, position of the joints, velocity of the joints, robot-ground contact features. We run 5 different seeds for each configuration and report the average return per epoch and the 95% CI.

This suggests that basing the design of the observation space from similar environments, something common in domain-specific literature, might lead to relying on suboptimal feature sets.

We argue that not defaulting to using as many features as possible also has tangible implications. In some applications like robotics, placing sensors that provide additional features might be costly, especially if done at scale (Alatise & Hancke, 2020). Therefore, doing so should only occur when the extra features provide a clear learning advantage. In addition, it could also be too intrusive for the human interacting with the RL agent (e.g., requiring information related to gender or race). This is an important issue for the broad ML community (Corbett-Davies & Goel, 2018) and better understanding tradeoffs between intrusiveness and learning outcomes is an ongoing effort.

## 2.2 LITERATURE NOT CONVERGING

We also observe that consensus on RL observation spaces does not naturally emerge in domain-specific communities. To further highlight this issue, we analyze the Traffic Signal Control (TSC) problem (Wei et al., 2021; Higuera et al., 2019; Zhou et al., 2019), which has also been a use case in other studies focusing on empirical aspects of RL design (Jayawardana et al., 2022). The TSC problem consists of controlling a set of traffic lights at a road intersection in order to optimize traffic flow. Even though there are more than 160 peer-reviewed studies proposing new RL approaches to solve the problem or aspects of it (Noaeen et al., 2022), and many high-fidelity simulators have been developed (Lopez et al., 2018; Zhang et al., 2019; Mei et al., 2022), the deployment of RL systems in real road networks is still a path to be traversed.

In Table 1 we present different observation spaces for the TSC problem found in the literature. Specifically, we look at the most recent papers identified in the systematic literature review from (Noaeen et al., 2022). The vast majority of the methods propose feature-based spaces encoding features from one or more of six different categories.

We can observe that there is little overlap between observation spaces considered in different works. In addition, multiple studies that use the same feature might encode it differently. As a result, when designing a new RL method that addresses the TSC problem, it is not clear which features it should rely on to get the best performance. This phenomenon is also observed in other applications such as robotics (Kim & Ha, 2021).

## 3 FEATURE SELECTION VIA RANDOM POLICY SAMPLING

The previous section underlies that there is room for improvement in current observation space design strategies for RL. The problem of identifying how many and which features to use has already been studied in RL (Liu et al., 2015), and different approaches have been proposed to do so (Bertsekas et al., 1988; Keller et al., 2006; Parr et al., 2008; Liu et al., 2012; Shen & Chi, 2016; Liu et al., 2021). However, current methods rely on trained agents to assess a certain feature set and thus often multiple training runs are needed. In the context of real-world problems, where environments are complex and training often consumes non-negligible time and resources, two problems arise: 1) training multiple times is costly, and 2) other design choices such as the RL algorithm can impact the outcome.

Table 1: Feature utilization in the observation space for multiple papers using RL for Traffic Signal Control. Feature categories include the current phase, the occupancy of each lane in the intersection, the position of the vehicles in the lane, the vehicles' speed, and their waiting time. In addition, some observation spaces are encoded as images of the intersection seen from above. Reference groups: (A) Jin & Ma (2019); Chu et al. (2019), (B) Wei et al. (2019b); Horsuwan & Aswakul (2019); Wei et al. (2019a); Zheng et al. (2019) (C) Kitagawa et al. (2019); Rizzo et al. (2019a); Aslani et al. (2019); Rizzo et al. (2019b) (D) Reda et al. (2019) (E) Shabestray & Abdulhai (2019) (F) Chen et al. (2019); Zhou et al. (2019) (G) Huang et al. (2019); Shu et al. (2019) (H) Gong et al. (2019) (I) Higuera et al. (2019) (J) Ge et al. (2019) (K) Kim & Jeong (2019).

| Reference group | Feature categories observed for each lane of the intersection | | | | | |
|---|---|---|---|---|---|---|
| | Phase | Occupancy | Position | Speed | Waiting time | Spatial encoding |
| (A) | | ✓ | | | ✓ | |
| (B) | ✓ | ✓ | | | | |
| (C) | | ✓ | | | | |
| (D) | ✓ | ✓ | ✓ | ✓ | | ✓ |
| (E) | | ✓ | ✓ | ✓ | | ✓ |
| (F) | | ✓ | | ✓ | | |
| (G) | ✓ | | ✓ | ✓ | | |
| (H) | | ✓ | | | | ✓ |
| (I) | | | | | ✓ | |
| (J) | | | ✓ | ✓ | | ✓ |
| (K) | | ✓ | | | | |

To address this gap, we propose to leverage inexpensive ensembles of random policies and multiple rollouts to identify statistically significant differences between candidate observation spaces. Simply put, when comparing two state representations, we propose collecting multiple rollouts from several random policies and generating a distribution of average returns for each representation. Then, we conduct a statistical analysis to determine whether one representation is more advantageous. This strategy does not require training the agent, thus avoiding the need to design and finetune other components of the RL setup, and reducing the amount of resources required.

While the concept of random search is not new in ML (Bergstra & Bengio, 2012), its potential in RL has not been sufficiently explored. Previous work demonstrates that random search provides more cost-efficient RL agents without substantially affecting performance (Mania et al., 2018). It can also be used as a good warm-starting approach (Barrett et al., 2020). We argue that combining random policy sampling with a search approach such as grid search or Hill Climbing (Russell & Norvig, 2002) can provide a good understanding of which features are relevant for a particular problem without incurring large computational costs.

This section explains our approach in more detail. We begin by building theoretical intuition into our method, using a simple use case with linear random policies. Then, we use a grid world example to practically illustrate our approach under a series of simple observation spaces. We conclude by integrating random policy sampling into the Hill Climbing algorithm, outlining the proposed feature selection method end-to-end.

### 3.1 THEORETICAL ANALYSIS OF RANDOM POLICY SAMPLING

We first present a simple theoretical example to justify our choice of random policy sample. We consider a toy environment that can be modeled by the tree shown in Figure 2a. Nodes in the tree are states, and two actions are possible: $A_1$ (left) and $A_2$ (right). We assume an observation space of one single feature $x$ and a linear policy of the form $f(x) = a \cdot x + b$. In all cases, if $f(x) < 0$, we take $A_1$, $A_2$ otherwise. The tree has a root node in yellow that acts as an initial state, and leaf nodes in which the reward is zero (gray), +1 (green), or -1 (red).

In this case, we want random policy sampling to guide us in determining whether feature $x$ matters, therefore we compare the case in which the policy takes the aforementioned form with the case in which the policy is simply $f(x) = b$. Random policy sampling provides us with random values for $a$ and $b$, which we assume follow $a, b \sim \text{Unif}[-1, 1]$. Given the policy is linear, for any $x$, the space of values for $a$ and $b$ can be partitioned with a straight line through zero (see Figure 2b). Therefore, randomly sampling $a$ and $b$ entails that $f(x)$ will have positive sign half of the times.

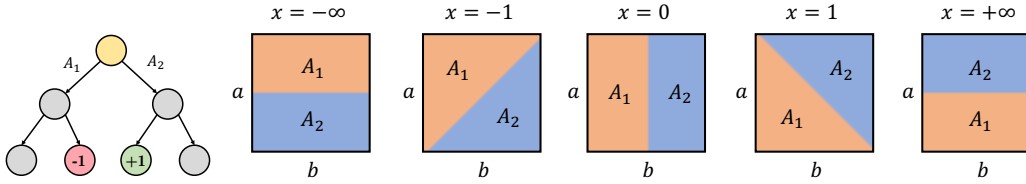

(a) State tree.  (b) Partition of the space of parameters $a$ and $b$ for different values for feature $x$.

Figure 2: Toy experiment in which randomly sampling $a$ and $b$ leads to finding policies with return +1 only when $x$ is considered.

One can realize that, in the absence of feature $x$, the bias-only policy consistently achieves zero reward, no matter the value of $b$. If constantly taking the same action led to the best reward, then no features would be needed. When considering the policy with feature $x$, for any pair of consecutive nodes, there exists a certain probability of taking the same action in both nodes or not; it depends on the overlap between the action-partition as depicted in Figure 2b for each value of $x$. Therefore, with enough rollouts, for some pairs $a$ and $b$, the best reward will be achieved. However, the opposite is also true, one can randomly sample policies that lead to the worst reward. Therefore, when collecting rollouts from multiple policies, we have to look at the right tail of the distribution.

While there exsist problem structures that are robust against this approach (e.g., when always taking the same action is the best), the majority of real-world problems follow a similar structure to Figure 2a's, in which there is a distribution of best actions depending on the observation space.

### 3.2 GRID WORLD EXAMPLE WITH RANDOM LINEAR POLICIES

To better understand random policy sampling, we design a toy experiment consisting of a grid world environment. At the beginning of every episode, the agent is placed on a random cell in the grid and its position is defined as $\boldsymbol{p_{a_0}} = (x_{a_0}, y_{a_0})$. Similarly, a different goal cell is randomly selected when the episode begins, its position being $\boldsymbol{p_g} = (x_g, y_g)$. We define the complete observation space of the problem as $\boldsymbol{s_t} = (x_g - x_{a_t}, y_g - y_{a_t})$; the agent can move up, down, left, and right as long as it remains inside the grid. Whenever the agent reaches the goal position, it receives a reward of 10 and the episode ends. If the agent reaches a certain timestep $t'$ without getting to the goal cell, it receives a reward of -10 and the episode also terminates. In any other case, the reward is zero.

Our goal is to conduct multiple rollouts in this environment using different random policies. To that end, we rely on a linear policy, defined as follows. At each timestep $t$, we compute $f_{1_t} = \boldsymbol{w_1} \cdot \boldsymbol{s_t}$ and $f_{2_t} = \boldsymbol{w_2} \cdot \boldsymbol{s_t}$, where $\boldsymbol{w_1}$ and $\boldsymbol{w_2}$ are linear coefficients sampled uniformly between -1 and 1 for each policy (including a bias term). To decide which action to take at each timestep, we look at the signs of $f_1$ and $f_2$ and assign one of the actions to each of the four possible sign combinations.

It is straightforward to see that an RL agent needs to have access to both features of the observation to learn the best policy; any other representation is suboptimal or overdefined. We show this in Figure 3. We define three alternative observation spaces by masking one or both features (e.g., one of the alternative spaces is $\boldsymbol{s_t} = (0, y_g - y_{a_t})$). Then, we use random policy sampling and collect $N_\pi$ sets of weights $\boldsymbol{w_1}$ and $\boldsymbol{w_2}$; we repeat this process for each of the state spaces $\mathcal{S}_i$. For each set of weights, we run $N_R$ rollouts and compute the average rollout return, resulting in a distribution $\mathcal{G}_{\mathcal{S}_i}$ for each space considered. As discussed in the previous section, we are only interested in the right tail of this distribution, therefore we only take the best 15% of average returns per space. Figures 3a - 3c compare the right tail of the return distribution when using both features (XY) against the cases in which only one feature is used (X or Y) or the observation is always zero (None).

In all cases, the distribution of returns corresponding to the observation space that includes both features achieves a longer tail and a higher average return. This indicates that, during the sampling process, there are certain pairs of weights $\boldsymbol{w_1}$ and $\boldsymbol{w_2}$ that, by relying on both features, lead to higher returns in this environment. This remark is aligned with the premise that the best observation space is given by both features. To further illustrate this, in Figure 3d we add a third feature: the number of steps until the episode terminates (i.e., the difference $t' - t$). By comparing the distribution for the case in which this feature is used (XYT) against the case in which it is masked (XY), we can see that the former adds artifacts that prevent random sampling from finding good sets of weights more often.

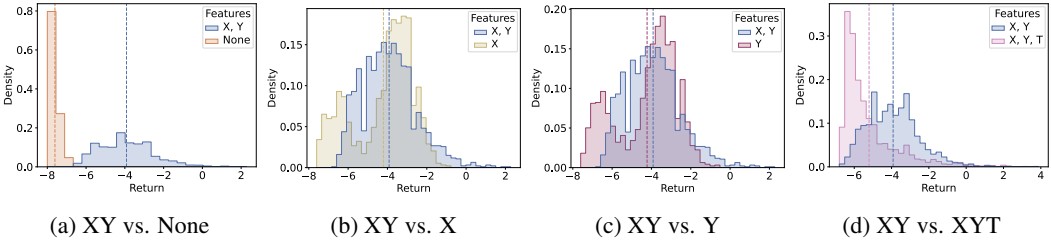

Figure 3: Distribution of average rollout returns when doing random policy sampling in the grid world environment. We compare the distributions obtained for the optimal observation space using both features XY against suboptimal spaces using (**a**) no features, (**b**) only feature X, and (**c**) only feature Y. We also compare the optimal space against a (**d**) overdefined observation space (XYT). We show the average of each distribution with a dashed line. In all cases, the optimal distribution achieves a longer tail and higher average returns.

### 3.3 INTEGRATION WITH HILL CLIMBING

Having established the utility of random policy sampling for evaluating observation spaces, we now combine it with the Hill Climbing (HC) algorithm to produce optimized representations. Our method couples the HC algorithm's iterative search with the cost-efficient evaluation of random policy sampling. For each iteration, the algorithm explores neighboring state space representations. For every neighbor, random policies are sampled and rollouts are conducted; the sampling process can be parallelized. A t-test (Walpole et al., 1993) is then performed to compare the resulting return distributions. Based on this statistical comparison, the algorithm navigates towards representations that show significant improvement. This iterative process stops once no statistically conclusive improvements are observed among neighbors. The process is described in Algorithm 1.

---

**Algorithm 1** Hill Climbing Integration with Random Policy Sampling

---

1: **Input:** initial_obs_space
2: **Output:** optimized_obs_space
3: $current\_obs\_space \leftarrow initial\_obs\_space$
4: **while** not converged **do**
5:     $neighbors \leftarrow$ generate_neighbors($current\_obs\_space$)
6:     **for** each $obs\_space$ in $neighbors$ **do**
7:         $policies \leftarrow$ sample_random_policies($obs\_space$)
8:         $returns \leftarrow$ conduct_rollouts($policies$)
9:         $avg\_return, p\_value \leftarrow$ t_test($returns, current\_obs\_space\_returns$)
10:     **end for**
11:     **if** a $obs\_space$ in $neighbors$ shows higher $avg\_return$ and $p\_value <$ significance_level **then**
12:         $current\_obs\_space \leftarrow obs\_space$
13:     **end if**
14: **end while**
15: **return** $current\_obs\_space$

---

The combination of HC and random policy sampling offers an approach to observation space design that is both systematic and cost-efficient. While the stochasticity of this approach does not guarantee always taking a step towards a better observation space, its inexpensive and straightforward nature makes it suitable for screening multiple candidate observation spaces in real-world problems where policy training requires a large computational overhead. Our method can be also used to identify a reduced set of promising observation spaces, which practitioners can further assess more efficiently.

## 4 EXPERIMENTS

We now apply random policy sampling and its integration with the HC algorithm in the context of two RL problems: a single-player version of the paddle game and the real-world problem of TSC, introduced in Section 2.2.

### 4.1 APPLYING HILL CLIMBING TO A PADDLE ENVIRONMENT

We first take the *Paddle* environment (Verma, 2020), a single-player version of the Pong game in which the goal is to hit, for as long as possible, a ball moving in a 2D space with a paddle that can only move in the left-right direction. The environment's full observation space consists of five features: the $x$ position of the paddle, the $x$ position of the ball, the $y$ position of the ball, the $x$ velocity of the ball, and the $y$ velocity of the ball. Our objective is to determine whether we can reduce the observation space for this problem applying the HC algorithm described in the previous section.

We take the environment configuration reported in the codebase and define a family of random policies consisting of a two-layer multilayer perceptron (MLP) in which all parameters are randomly sampled following uniform distributions (full experiment configuration reported in Appendix B). For each space, we sample multiple policies and compute the average rollout return for every policy. Within the distribution of average returns, we take the right tail for each observation space. There are 32 possible observation spaces for this environment; since the outcome of the HC algorithm depends on its initialization, we repeat this experiment using each possible observation space as initial space and report statistics over the outcomes; this is summarized in Table 2.

We report the observation spaces that, for one or more of the 32 initializations, have been returned by the HC algorithm. We also include the average number of spaces evaluated per trial. In the baseline scenario we can see there are different candidate observation spaces. In all cases, the number of evaluated spaces is less than one third of all the possibilities. We now train an RL agent using each of the observation spaces in Table 2 and report its performance in Figure 4a. The returns show that, among the candidate observation spaces, the one that uses all features reaches the best outcome. In all cases, we train using the same neural network structure used during random policy sampling.

Table 2: Running the Hill Climbing algorithm in the paddle environment with all possible observation spaces as initial space. We report the observation spaces that achieved the best result and how many times they did (out of 32). In addition, we include the average number of spaces evaluated (max. is 32). Observation spaces are reported as binary strings indicating whether feature in position $i$ was used.

| Baseline scenario | | | Hard scenario | | |
|---|---|---|---|---|---|
| **Obs. space** | **Successes** | **Avg. num. evals.** | **Obs. space** | **Successes** | **Avg. num. evals.** |
| 00111 | 15 | 12.9 | 00111 | 12 | 12.4 |
| 11111 | 12 | 10.7 | 10011 | 12 | 11.4 |
| 11010 | 3 | 8.7 | 11111 | 4 | 9.8 |
| 10001 | 2 | 8.0 | 01011 | 4 | 9.8 |

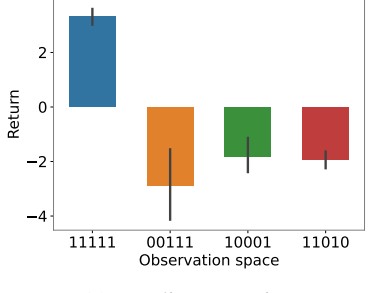

(a) Baseline scenario.

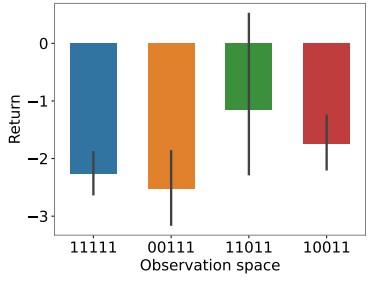

(b) Hard scenario.

Figure 4: Training RL agents using each of the candidate observation spaces. Observation spaces are reported as binary strings indicating whether feature in position $i$ was used. 95% CI is shown.

To further assess the utility of our approach, we define a second configuration for the paddle environment in which the ball and the paddle move faster, making the task harder. We repeat the same procedure and report the results in Table 2 (Hard scenario) and Figure 4b. When the task becomes harder, using all features does not confer the best advantage. Instead, another of the proposed observation spaces achieves better return. These results confirm the basis of our approach: trading optimality for search cost. We are able to identify good candidate state spaces, which sometimes involve using less features, without the need to train RL agents during exploration.

## 4.2 RANDOM POLICY SAMPLING FOR TRAFFIC SIGNAL CONTROL

In this section we test our approach using a real-world problem, TSC. The goal is to control the traffic lights at an intersection by setting them to different phases. A phase is defined as an assignment of states to each traffic light (e.g., green, red). The goal is to optimize the traffic flow at the intersection by performance metrics such as the average waiting time. This problem can be formulated in multiple ways that affect the specific available actions, the timescale, and the concrete goals to achieve (Eom & Kim, 2020). In this work we focus on the most frequent formulation, in which time is discretized in equal intervals and a phase from a set of predefined phases is chosen at each timestep.

We carry out several experiments to gain insights on the significance of the features in Table 1. To that end, we leverage the RESCO benchmark for TSC (Ault & Sharon, 2021) and a real-world road network and simulation with 21 agent-controlled traffic lights as our scenario. Among those, we control seven of them with RL (blue dots in Figure 5a).

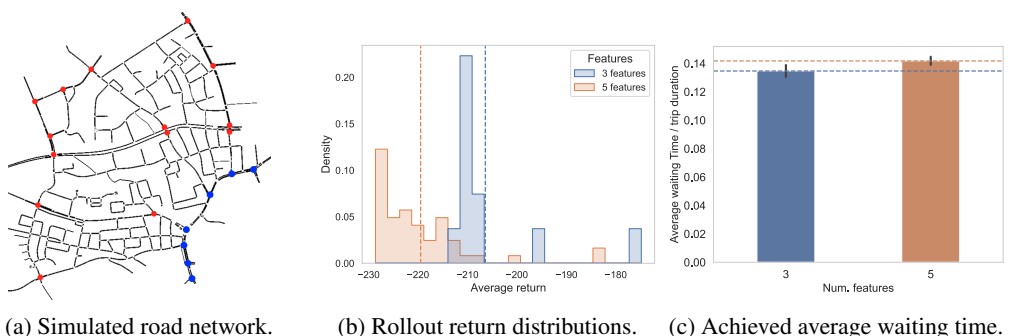

(a) Simulated road network.      (b) Rollout return distributions.      (c) Achieved average waiting time.

Figure 5: Experiments in the Traffic Signal Control problem using the RESCO benchmark. (**a**) We simulate a real-world road network with 7 RL-controlled intersections (blue dots). (**b**) After running the Hill Climbing algorithm with random policy sampling, we find a reduced observation space with just 3 features that, compared to the full observation space with 5 features, achieves higher average rollout return in the right tail of the distribution. (**c**) After training an RL agent with both observation spaces, the reduced one achieves lower (better) average waiting time per vehicle. 95% CI is shown.

We take the observation space used in one of the benchmark models in RESCO; it is composed of five different features for each lane in the intersection (an intersection has a variable number of lanes): the current phase, the number of vehicles approaching, the average waiting time, the number of vehicles waiting, and the average vehicle speed in the lane. After running the HC algorithm, an alternative space with only three features (the first, third, and fourth) is proposed. The distribution of rollout returns in the right tail is shown in Figure 5b for the full observation space (5 features) and for the reduced one (3 features). We use 500 random policies in each case.

Finally, we take both the full observation space and the reduced space with three features and train an RL agent for each using the default configuration in RESCO. This corresponds to setting the action to be the phase at each timestep and the reward is the total waited time at an intersection multiplied by -1; additional implementation details can be found in Appendix C. As shown in Figure 5c, the proposed space with three features achieves lower (better) waiting time per vehicle compared to using all features. These results validate the usefulness of our approach in real-world RL problems. This is especially relevant in environments in which training is costly.

## 5 RELATED WORK

**Design of observation spaces for real-world problems**    Our work aligns with improving the design of observation spaces for practical applications. Singh et al. (2020) discuss what makes a good state representation for an RL task and outline why it is a nontrivial problem. Reda et al. (2020) emphasize that the observation space design process in many practical applications is empirical and benchmark different observation spaces for robot locomotion. Kim & Ha (2021) extend the analysis of observation spaces for robotics and propose a method to select features as the agent trains. Our work is also framed in the context of practical applications and empirical analysis; we propose a cost-efficient method to select raw features that complements empirical design in real-world tasks.

**Learning state representations and abstractions**    Other works don't optimize over the space of raw features; instead they use all features and focus on learning abstractions that improve agent performance (Li et al., 2006). Different approaches stand out: factoring and reducing MDPs (Givan et al., 2003), randomly sampling features at each iteration (Afshar et al., 2020), using contrastive learning for learning Markov state abstractions (Allen et al., 2021), learning invariant representations (Zhang et al., 2020), treating the state as an evolvable entity following a curriculum (Wang et al., 2019), choosing different transformations for image-based states (Raileanu et al., 2020), and using canonical states to remove redundancy across observations (Wu et al., 2017). In this paper we focus on optimizing the raw state representation instead of relying on all the feature set to create an abstraction from it; we seek to minimize the number of sensors required in the real setup. Our work can be seen as a previous step before applying some of the methods above.

**Feature Selection in Reinforcement Learning**    The problem of selecting features has been long-studied in ML, and the number of studies addressing it does not cease to grow (Hall, 1999; Chandrashekar & Sahin, 2014; Cai et al., 2018). While the majority of works are framed within supervised learning, there are studies which have specifically addressed feature selection and dimensionality reduction in RL (Liu et al., 2015). Different methods that have been proposed include aggregating states when approximating value functions (Bertsekas et al., 1988; Keller et al., 2006; Parr et al., 2007; 2008), mutual information (Hachiya & Sugiyama, 2010; Shen & Chi, 2016), regularization (Loth et al., 2007; Kolter & Ng, 2009; Liu et al., 2012; Hao et al., 2021), matching pursuit methods (Johns & Mahadevan, 2009; Painter-Wakefield & Parr, 2012), entropy-based dimensionality reduction (Tangkaratt et al., 2016; Parisi et al., 2017), and using RL itself to pick features (Liu et al., 2021). The majority of this methods are approximate, require several training runs, and have not been studied in the context of real-world problems. We propose a straightforward and cost-efficient method that does not require training and which can be applied to multiple practical applications.

**Practical considerations of empirical RL**    Our work is framed within the study of empirical RL and the practical problems of RL. Progress in RL is brittle; the real world is challenging (Dulac-Arnold et al., 2021) and current RL practices have been proven not to be robust against different elements such as experimental setups (Henderson et al., 2018), codebases (Engstrom et al., 2019), hyperparameter tuning (Islam et al., 2017; Zhang et al., 2021), and statistical reporting (Agarwal et al., 2021; Colas et al., 2018; 2019). Design choices have a major impact in practical RL; Reda et al. (2020); Kim & Ha (2021) consider the effect of multiple design choices on the agent side and Andrychowicz et al. (2020) run a similar analysis on algorithmic components. The effect of RL components on the performance vs. generality tradeoff is discussed by Hessel et al. (2019). Ceron & Castro (2021) find that smaller-sized environments are well suited for empirical work in RL, Jayawardana et al. (2022) study the impact of evaluating in MDP instances instead of MDP families, and Chan et al. (2019); Jordan et al. (2020) address the reliability of RL results. We further investigate the choice of the observation space and propose a method to support practical decision-making which does not need multiple costly evaluations.

## 6    CONCLUSION AND FUTURE WORK

In this paper, we focus on observation space design in real-world reinforcement learning (RL). We propose an inexpensive method to gauge the relevance of state features based on random policy sampling, which does not require training. This is an overlooked problem in the literature, since the structure of RL renders many feature selection methods designed for supervised learning of little use and the proposed solutions require training multiple times. We demonstrate that random policy sampling, combined with a search algorithm such as Hill Climbing, is well-suited to find good observation spaces in a simple way that can be leveraged in practical applications. We evaluate our approach in different RL problems, including Traffic Signal Control, which has been attempted many times using RL and feature selection still remains one of its unsolved aspects. While this work focused on feature selection without training at all, future work will consider combining this random search strategy with limited training paradigms, such as transfer learning between different observation configurations. Overall, we hope our findings help gain momentum for the study of how RL state spaces should be designed in practical contexts and contribute to a better understanding of the brittleness in RL research.

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
