# OpenReview forum: "Look Ma, No Training! Observation Space Design for Reinforcement Learning"
_ICLR.cc/2024/Conference — ICLR 2024 Conference Withdrawn Submission_

### Official Review · Reviewer_N2rj · 2023-10-22

**Soundness:** 2 fair
**Presentation:** 3 good
**Contribution:** 3 good
**Rating:** 5
**Confidence:** 4

**Summary:**

The authors propose a new method for constructing the observation spaces of reinforcement learning environments. They propose to use the distribution of returns of policies with random weights to evaluate an observation space by examining the right tail of the distribution. This has the significant advantage over the more naive approach of training multiple agents that it only requires evaluation and so observation spaces can be constructed with fewer environment samples. The authors combine examining the right tail of the return distribution with hill-climbing to automatically find good observation spaces.

They conduct experiments on paddle and traffic signalling as well as a motivating example on pybullet environments.

**Strengths:**

The paper has some significant strengths:
- The method proposed is novel and interesting. In particular designing observation spaces without training is a very nice property of their approach.
- The paper is very clear and generally well written. The toy examples given that motivate their approach are clear and interesting.

**Weaknesses:**

Despite its clear strengths, the paper also has some notable weaknesses.

My major concern is that I am not convinced they have adequately demonstrated that using too many features significantly harms performance. In Figure 1, the line with no ablation (in blue) seems to me to pretty much on par with the other approaches, with leaving out features making little difference, if any.

Again, in the Paddle experiment, having all of the features gives the best result in one of the scenarios, and clearly has a lower return than leaving out the x- and y-position of the ball, but the error bars are large. It is also not clear to me why leaving out the x- and y-position of the ball would be beneficial in this environment. It seems very counter-intuitive for the performance of the agent to improve by not knowing where the ball is -- the only reason I can think of for this is that the position of the ball is not sufficiently stochastic to be required, but then that suggests a problem with the environment, rather than genuinely discovering a feature that should be left out.

Perhaps the most convincing case for this problem is made by the traffic signal results, but there the gains also seem marginal.

Another issue is that the authors only seem to cover environments with a small number of features. It is not clear to me that their approach obviously scales to environments with a large number of features.

Finally, a small issue is that this approach only works for hand-engineered observation spaces and not large-dimensional observation spaces such as image-based ones.

**Questions:**

- Have you tried running experiments on environments with more features?
- How stochastic are the environments that you experiment on? What sources of stochasticity are there? This is a very important consideration because in a fully deterministic environment, all features are useless -- they just might help with the search problem.

---

### Official Review · Reviewer_NkaL · 2023-10-22

**Soundness:** 1 poor
**Presentation:** 3 good
**Contribution:** 2 fair
**Rating:** 1
**Confidence:** 4

**Summary:**

The paper describes an approach to perform feature selection for the observation space for RL algortihms without having to train an agent. The approach rolls out random policies with a selected subset of the features (which forms the observation space) in a given environment and takes the right tail of the performance distribution to determine if the used feature set is performing well. It initialises with a certain observation space and then generates and evaluates neighbours of the observation space until no improvement is achieved over the current observation space (Hill Climbing). It uses a t-test to determine if a neighbour is statistically significantly better than the current observation space. It shows the usefulness of random policy rollouts for determining well-performing observation spaces for the PPO algorithm on some PyBullet environments. The paper evaluates the proposed approach on a paddle game similar to Pong and a Traffic Signal Control (TSC) environment where 7 out of 21 traffic lights are controlled with an observation space of up to 5 features.

**Strengths:**

The approach seems novel and it has a fairly good chance of working in principle. The problem the paper is trying to tackle is also a farily important one in my opinion and there needs to be more research in this direction. The clarity of the writing is good in general.

**Weaknesses:**

**Experiments**:
The main weakness for me is that not enough environments or types of features were evaluated on, nor were enough baselines compared to. The experiments done seem fairly cheap (though neither the amount of compute used nor the infrastructure used is provided as far as I can see), so I do not know why the main experiments were done on only two small environments.
Regarding there being no other baselines, one simple baseline to compare against could be to train an RL agent for some time with a subset of features and see how it performs. I would not be surprised if this baseline were cheap as well. In random policies, a lot of compute is wasted on bad random policies. Without a comparison of compute used in both kinds of approaches, we cannot really judge if this method is actually cheap.
I also did not understand why other types or kinds of features were not used during the evaluation. Using just two examples with five features each is not very convincing for me. Neither is the performance gained so clear. In fig. 4b for example, the confidence bars are so large that it is hard to trust the results.
It is also not clear how one would avoid local optima in nearest neighbour hill-climbing approach proposed. This would easily break down probably in even simple environments in my opinion.
How was the hyperparameter of the percentage of the right tail distribution used to evaluate the random policies chosen? This could easily be chosen to get better results for the approach and is important to know.
For the Traffic Signal Control experiments, why are their convolutional layers used (see Table 7, Appendix) when the agent state seems to be feature-based?

**Writing**:
Several important details seem to be missing from the paper.
What is the reward sturcture of the Paddle environment?
How are the actions in section 3.2 chosen in the case of an over-/under-defined feature space? (for the "normal" feature space of 2 features it is clear how the signs lead to 4 actions, but not for the other feature spaces).
Even though section 3.1 is about theoretical analysis, I do not see much theory there. It feels more like using a narrow and rhetorical example to motivate the approach.
The RL agent trained in section 4.2 is not mentioned in the main paper.

**Questions:**

Please see the weaknesses above. Apart from the explicit questions there, why were there not more environments, baselines and types of feature sets experimented upon? How would one avoid local optima in the approach?

---

### Official Review · Reviewer_fGnS · 2023-10-26

**Soundness:** 2 fair
**Presentation:** 3 good
**Contribution:** 1 poor
**Rating:** 5
**Confidence:** 5

**Summary:**

In this paper, the authors propose to do random rollouts which use different observation spaces so that it will be easier to find out which observation space works the best for a specific task.

**Strengths:**

1. The paper is well written and easy to understand.

2. The paper provides an algorithm easy to implement and it should be reproducible with what is described in the paper.

**Weaknesses:**

1 . The intuition of the paper is not entirely justified.

In general I believe people avoid reward engineering in research since it goes against what deep RL promises for real-life problems.
The idea of using neural networks is that we don’t want to touch the reward space and instead let the neural networks figure it out by itself.

I agree for one specific problem, reward engineering could make training faster.

For example input normalization is the most common reward engineering that is being used.

I would argue there are some issues for this idea:
a) Reward engineering is not scalable for more complex problems.

Real robotics problems are usually in image space or using lidar, 3D point cloud or some combination of them.

And it is not clear how the proposed method could be used or how much performance gain it would introduce.

b) Engineering in reward might not be worth it compared to engineering in optimization.

Instead of doing rollouts in reward space, a probably more efficient way is to do “rollouts” or population based training in how the network is optimized.

[1] has been proved to be quite engineeringly efficient.

And It would be interesting to see if given the same computation budget, how the two methods compared.



2 . The experiment sections do not include complex RL benchmarks.

[1] Jaderberg M, Dalibard V, Osindero S, Czarnecki WM, Donahue J, Razavi A, Vinyals O, Green T, Dunning I, Simonyan K, Fernando C. Population based training of neural networks. arXiv preprint arXiv:1711.09846. 2017 Nov 27.

**Questions:**

1. Does the final performance changes for different reward spaces if the network is trained for long enough?

---

### Official Review · Reviewer_CJ7V · 2023-10-30

**Soundness:** 1 poor
**Presentation:** 3 good
**Contribution:** 2 fair
**Rating:** 3
**Confidence:** 4

**Summary:**

The work proposes a novel observation space design algorithm based on random policy evaluation and hill climbing. The algorithm is applied to toy as well as more simulated real-world environments to demonstrate the ability to select sufficient sub-spaces.

**Strengths:**

The work addresses an important problem for RL and should be of relevance for a larger target audience at ICLR. A more principled way of doing observation space design could be beneficial for a variety of target domains and could have a significant impact in RL.

While the idea of probing environments with policies is not entirely novel (see, e.g., “Environment Probing Interaction Policies” or “Context is everything”), to the best of my knowledge, it is usually done to identify environment dynamics and learn policies that generalize across different dynamics and not to refine observation spaces.
Overall I find the idea compelling to use environment interactions to refine the observation space. Compared to learning observation space embeddings, the proposed way of “pruning” the observation space is quite interpretable while still preserving the original features.

**Weaknesses:**

The experiments do not seem to support the claims.

If I understand Table 2 correctly, then the Observation space “00111” was deemed the most suited space for the baseline task in the paddle game. However, Figure 4a) shows clearly that, out of the four spaces considered, “00111” is the worst possible space. The default space “11111” as well as the two other spaces found with HC “11010” & “10001” both outperform the supposedly best space.

For the hard scenario, the table and Figure 4b) are mismatched as the “best space” “11011” does not appear in the table. The two best spaces suggested by HC “00111” and “10011” both are accounted for in the plot and both perform much worse than “11011”, which does not appear in the table.

These results strongly suggest to me that HC is not a good candidate for selecting the observation spaces as in both scenarios, it did not suggest the best space reliably (or at all).

Further, the experimental setup in 4.2 makes me wonder why the proposed method was able to propose a pruned space that performed better than the original observation space with 5 features. Many details are missing here, such as how many HC runs were performed (this is important since you pointed out that HC strongly depends on the initialization) and how was the final space selected? How were the other hyperparameters for your method set? Why is it crucial to apply a right tail filter of top 5% here, whereas you previously used 15% for the experiment in section 3.2?

One big drawback of the method, which is not addressed in the work, is that the decoupling of the observation space selection completely ignores the learning dynamics of an RL agent. I believe that the method does not perform as well as it could, as it abstracts away any learning of an RL agent and thus might not see that certain features are important for learning at different stages of an agent’s life cycle.

Lastly, the work often highlights the importance of observation space design without the need for expensive training as this would be prohibitive in real-world scenarios. While I appreciate the sentiment, I do not see how random policies are more beneficial in this case. Take for example the TLC scenario. In this scenario the observation space selection was only possible because of a simulator. Application to real-world would still not be possible as you would not want to first run 500 random policies on your traffic lights to figure out which features are relevant. Thus, I feel the work is not well aligned with real-world applications. In particular, I wonder if the observation space design algorithm presented here might overfit to the simulation environment and abstract away important features that would make a policy transferrable from simulation to a real environment.

Overall, while I believe this is an interesting work, addressing an important challenge in RL, I do not think that the work is ready for publication.

**Questions:**

* How did you tune the hyperparameters of your method?
* How crucial are the following hyperparameters of your method listed in Appendix C?
    * #random policies
    * rollouts per policy
    * right tail filter
* Can you explain the discrepancy between Table 2 results and Figure 4?
* Will the code be made publicly available?